# DEMYSTIFYING LEARNING OF UNSUPERVISED NEURAL MACHINE TRANSLATION

## ABSTRACT

Unsupervised Neural Machine Translation or UNMT has received great attention in recent years. Though tremendous empirical improvements have been achieved, there still lacks theory-oriented investigation and thus some fundamental questions like *why* certain training protocol can work or not under *what* circumstances have not yet been well understood. This paper attempts to provide theoretical insights for the above questions. Specifically, following the methodology of comparative study, we leverage two perspectives, i) *marginal likelihood maximization* and ii) *mutual information* from information theory, to understand the different learning effects from the standard training protocol and its variants. Our detailed analyses reveal several critical conditions for the successful training of UNMT.

## 1 INTRODUCTION

Unsupervised Neural Machine Translation or UNMT have grown from its infancy (Artetxe et al., 2018; Lample et al., 2018a) to *close-to-supervised* performance recently on some translation scenarios (Lample & Conneau, 2019; Song et al., 2019). Early UNMT works (Artetxe et al., 2017; Lample et al., 2018a; Yang et al., 2018) adopt complex training strategies including model initialization, synthetic parallel data for warming up the model, adversarial loss for making encoder universal, different weight sharing mechanisms etc. Then Lample et al. (2018b) simplifies all these and establishes a two-components framework, involving an initialization strategy followed by iterative training on two tasks, i.e. **denoising auto-encoding** with the **DAE loss** and **online back-translation** with the **BT loss**. Works afterwards mainly focus on developing better initialization strategies (Lample & Conneau, 2019; Ren et al., 2019; Song et al., 2019; Liu et al., 2020). Although obtaining impressive performance, it is unclear *why* this standard training protocol is possible to be successful. Kim et al. (2020) and Marchisio et al. (2020) consider the standard training as a black-box and empirically analyze its success or failure regarding different data settings (i.e. text domains and language pairs). Unfortunately, due to the lack of theoretical guidelines, some fundamental questions are still remained unknown: *what* standard training tries to minimize under the general unsupervised training paradigm (Ghahramani, 2004) and *when* a certain training protocol can work for training UNMT? In this paper, we attempt to open the back-box training of UNMT and understand its theoretical essence from two angles: **i)** a *marginal likelihood maximization* view; and **ii)** an *information-theoretic* view by ablating standard training protocol with other variants. Our contributions are as follows.

**A.** By making an *analogy* of standard training protocol with *marginal likelihood* or *Evidence Lower BOund (ELBO)* optimization, we visualize the learning curves of the two terms in ELBO objective, and found that optimizing ELBO is not *sufficient* for training a successful UNMT model, indicating that specific regularization design i.e. the DAE loss, quite matters.

**B.** By leveraging information theory, we present a formal definition on what does it mean to *successfully* train an UNMT model, and then readily derive a *sufficient condition* and a *necessary condition* for successfully training UNMT in principle. In addition, we validate both sufficient and necessary conditions through empirical experiments, and find that both conditions indeed explain *why* standard training protocol works while others suffer from degeneration to learning sub-optimal tasks.

**C.** Based on explanations for those failed protocols, we continue experiments to settle the role played by DAE and BT. Firstly, BT is the *main* task while DAE is a critical *auxiliary*. Then we clarify that DAE has more important role than just learning word order, accepted as common knowledge in almost all previous works, but also *preserving* the mutual information between encoder input and

encoder output, which is *necessary* for successful training. Furthermore, DAE also functions as a behavior regularizer for decoding with online BT, and prevents BT from yielding degenerated data.

## 2 UNDERSTANDING UNMT FROM TWO PERSPECTIVES

In this section, we first introduce background about the standard training protocol proposed in Lample et al. (2018b), which is adopted by almost all later works. Then we introduce the basic concept of two perspectives on which we rely for analyzing the learning of different training protocol variants. Due to the space limit, please refer to appendix A.1 for a timely literature review of recent advance.

### 2.1 STANDARD TRAINING PROTOCOL

The standard training protocol involves standard *initialization strategy* and standard *iterative training procedure*, and they both are built upon a specific design of encoder-decoder *parameterization*.

**Parameterization and initialization** UNMT model adopts a shared embedding matrix for a shared vocabulary with joint BPE (Sennrich et al., 2016), and the two languages share the same encoder and decoder with only a language embedding for distinguishing the input from different languages. As a result, unconstrained decoding might generate tokens from the same language as the input. Standard initialization means using fastTEXT (Bojanowski et al., 2017) to initialize the embedding matrix, denoted as *JointEmb*. *XLM* (Lample & Conneau, 2019) uses a trained encoder to initialize both encoder and decoder of the UNMT model. We also consider *random* initialization for completeness.

**Iterative training strategy** The iterative training strategy involves optimization of two critical losses *by turns*, i.e. the DAE loss and the BT loss as defined in Eq. 1 and Eq. 2, where $s$ and $t$ denote the two languages. DAE loss is constructed through sampling a monolingual sentence $x$ (or $y$), construct its noisy version $C(x)$ ($C(y)$) and minimize the reconstruction error or RecErr:

$$\mathcal{L}^{dae} = -\log p_{s\to s}(x|C(x)) + [-\log p_{t\to t}(y|C(y))], \tag{1}$$

BT loss is constructed through sampling a monolingual sentence $x$ (or $y$), construct its corresponding translation via the current model $\mathcal{M}(x)$ ($\mathcal{M}(y)$) through back-translation and minimize the RecErr:

$$\mathcal{L}^{bt} = \mathbb{E}_{\hat{y}\sim\mathcal{M}(x)}[-\log p_{t\to s}(x|\hat{y})] + \mathbb{E}_{\hat{x}\sim\mathcal{M}(y)}[-\log p_{s\to t}(y|\hat{x})], \tag{2}$$

The online BT process involved in the iterative training strategy can be seen as Co-Training (Blum & Mitchell, 1998), where two models (*with shared weights*) constructed on two views (*source/target sentence*) generate pseudo labels as the other view (*pseudo translation*) for training the corresponding dual model. We summarize the whole standard training protocol in Algorithm 1 in appendix A.2.

**Constrained decoding** Besides the basics, we further introduce the concept of *constrained decoding* where the model should be constrained to decode tokens only in the *target* language regardless of the shared embedding parameterization. This could give us a simple definition of **cross-lingual** RecErr beyond naive RecErr in Eq. 2. Details of the algorithm and the definition are shown in appendix A.3.

### 2.2 A MARGINAL MAXIMIZATION VIEW

The standard training of UNMT model takes advantage of sole monolingual corpora $\mathcal{D}_s$, $\mathcal{D}_t$, which is similar to the generative modeling setting where only unlabeled data is available (Ghahramani, 2004). Here we take an analogy of the standard UNMT training as *implicitly* maximizing marginal of the monolingual data. Due to the duality of translation (He et al., 2016), the target sentence not only plays the role of label, but also the input in reverse translation direction. So in essence the standard UNMT training can be seen as maximizing the marginal log likelihood of $\mathcal{D}_s$ and $\mathcal{D}_t$ simultaneously. However, since marginals involve infinite summation over a certain view (target/source), a lower bound is often optimized via Monte Carlo approximation (Kingma & Welling, 2014).

In the following derivation of ELBO (Kingma & Welling, 2019), $q_\phi(y|x)$ is the posterior distribution of $y$ when taking $y$ as the latent variable. Here we only derive the bound for $x \in D_s$. A detailed analogy of the standard UNMT objective and the ELBO objective is presented in Table 1. As you can see, both objectives have the same *reconstruction error* terms but different *regularization* terms: for ELBO, the model is optimized to stay close with the language model via the KL loss.

Table 1: An anology of the standard UNMT objective and the negative ELBO objective. Note that both objectives have the reconstruction term; since in VAE training KL divergence is often referred to as regularization (Prokhorov et al., 2019), so we call DAE loss the regularization term as well.

|  | *reconstruction error* | *regularization term* |
|---|---|---|
| **standard UNMT** | $-\mathbb{E}_{\hat{y}\sim\mathcal{M}(x)}\log p_{t\to s}(x|\hat{y})$ | $-\log p_{s\to s}(x|C(x))$ |
| **negative ELBO** | $-\mathbb{E}_{q_\phi(y|x)}\log p_\theta(x|y)$ | $D_{KL}(q_\phi(y|x)||p_\theta(y))$ |

$$\log p_\theta(x) \geq \mathbb{E}_{q_\phi(y|x)}\Big[\log p_\theta(x|y)\Big] - D_{KL}(q_\phi(y|x)||p_\theta(y)) \quad \textbf{(ELBO)} \tag{3}$$

Worth noting that, we are not the first to make a connection between marginal maximization and the standard UNMT training. In He et al. (2020), they have already proposed an ELBO formulation for unsupervised sequence transduction task. However, they tend to focus on replacing the standard UNMT-training-style objective function with the ELBO objective, and propose several critical tricks such as *Gumbel softmax* and *self-reconstruction* for making ELBO really work. Instead, we leverage ELBO mainly as an *anology* to the standard UNMT training objective, and through comparative study with other protocol variants, we can further understand why standard objective and its variants work or not, even though they all tend to have *similar* ELBO values. Details are in appendix A.4.

## 2.3 AN INFORMATION-THEORETIC VIEW

If we denote $Y' = \mathcal{M}(X)$ as the random variable (*r.v.*) generated by the model $\mathcal{M}$ over $X$. Therefore, if $(Y', X)$ gradually contains more and more bilingual knowledge, the model will be able to generate better translations, eventually leading to the success of UNMT training. Suppose $c$ is a constant predefined by users which controls the satisfactory level for translation performance, we give a definition to formalize the *success* of UNMT training from an information-theoretic viewpoint.

**Definition 2.1.** If $\mathbb{I}(Y', X) > c$ after training, we say that UNMT training is successful; otherwise, we say that UNMT training fails. *(Caveat, c is concetual quantity, we never instantiate its value.)*

Suppose $p(x, y')$ is the true distribution of $\langle x, y' \rangle$, and $p_{t\to s}(x \mid y')$ is an estimator of $p(x \mid y')$. We obtain the following two conditions for success of training UNMT based on the above definition 2.1.

**Proposition 1.** (*Sufficient condition*) If $\mathbb{E}_{p(x,y')}\log p_{t\to s}(x|y') \geq c - \mathbb{H}(X)$, then UNMT training will be successful.

*Proof.* Based on Definition 2.1 and the definition of mutual information (MI) and Jensen's inequality, we can derive the following inequality (Pimentel et al., 2020):

$$\mathbb{I}(X, Y') = \mathbb{H}(X) - \mathbb{H}(X \mid Y') \geq$$
$$\mathbb{H}(X) - \mathbb{H}_{p_{t\to s}}(X \mid Y') = \mathbb{H}(X) + \mathbb{E}_{p(x,y')}\log p_{t\to s}(x \mid y') \geq c \tag{4}$$
$$\square$$

Since the sufficient condition relies on true distribution $p(x, y')$ which is unknown in practice, we sample $(x, y')$ from the empirical distribution of $X$ and $p_{s\to t}(y'|x)$ as approximation. Then the ideal sufficient condition is reduced to a *practical* one: If $\sum_x \mathbb{E}_{p_{s\to t}(y'|x)}\log p_{t\to s}(x|y') \geq c - \mathbb{H}(X)$, then UNMT training will be successful. Since MI is symmetric, we can also have similar formula regarding $s \to t$ direction with $\mathbb{E}_{p(x',y)}\log p_{s\to t}(y|x') \geq c - \mathbb{H}(Y)$. They together connect success of training to the BT loss in Eq. 2: *a lower BT loss is more likely to make UNMT training successful.*

Furthermore, if we denote the encoder output as a *r.v.*, $Z = enc(X)$, we can obtain the following *necessary* condition:

**Proposition 2.** (*Necessary condition*). If UNMT training is successful, then $\mathbb{I}(X, Z) \geq c$.

*Proof.* Following the *Data Processing Inequality* (Cover & Thomas, 1991), we have the following inequality that holds all the time:

$$\mathbb{I}(X, Z) \geq \mathbb{I}(X, Y') \geq c, \tag{5}$$

with the *Markov* chain (or data processing order) $X \xrightarrow{enc} Z \xrightarrow{dec} Y'$. $\qquad\qquad\square$

In subsequent experiments, we follow Pimentel et al. (2020) and estimate $\mathbb{I}(X, Z) = \mathbb{H}(X) - \mathbb{H}(X|Z)$ by calculating $\mathbb{H}(X)$ through a statistical 1-gram language model and $\mathbb{H}(X|Z)$ through probing (Conneau et al., 2018) respectively. For estimating $\mathbb{I}(X, Y')$, we use the *token-by-token* point-wise mutual information (PMI) over some pseudo bitext as a surrogate to the *sentence-by-sentence* MI. The detailed estimation methods are presented in appendix A.5.

## 3 EXPERIMENT

Table 2: Variants of training protocol.

| Protocol | Loss used |
|----------|-----------|
| **standard** | DAE (Eq. 1) + BT (Eq. 2) |
| **dae-only** | DAE (Eq. 1) |
| **bt-only** | BT (Eq. 2) |
| **elbo** | ELBO (Eq. 3) |
| **elbo-dae** | ELBO (Eq. 3) + DAE (Eq. 1) |

In this section, we conduct a series of carefully-designed comparative studies of the standard UNMT training protocol and its variants as well as using the ELBO objective. Different training protocols that we investigate and their name abbreviations are listed in Table 2. The logic flow of this section could be summarized as follows. **i)** Firstly, we present the basic experimental settings, the overall performance of carefully-designed ablations, and some highlighted observations from those results; **ii)** as promised in Sec. 2.2, we visualize the learning curves of different terms in *negative* ELBO with a conclusion that marginal maximization is only a *necessary* but *not sufficient* condition for successfully learning translation; **iii)** we explain both quantitatively and qualitatively why **dae-only** and **bt-only** that implicitly optimizes ELBO cannot work from an information-theoretic perspective, and highlight the importance of task-specific regularization such as the DAE loss; **iv)** we clarify the main and auxiliary relationship between BT and DAE loss, and further investigate the critical regularization effects of DAE loss.

### 3.1 EXPERIMENTAL SETTINGS AND OVERALL PERFORMANCE

**Dataset and Reproducibility** We adopt the publically accessible WMT14 En-Fr and En-De datasets for our experiments. We strictly follow the data pre-processing pipeline and instructions for training in the official XLM code repository [1]. The monolingual data for each language is set to about 5M sentences from the *newscrawl* monolingual collection [2]. Though we do not focus on improving over state of the art, adding more monolingual data can indeed largely improve the final performance. For XLM initialization, we download the pretrained models from the XLM repo; and for JointEmb initialization, we use fastTEXT (Bojanowski et al., 2017) for learning the word embeddings on the concatenated monolingual corpora for each language pair (about 10M).

**Outline of Overall Performance** Here in Table 3, we first report the overall performances with the **standard** training protocol and two of its variants **dae-only** and **bt-only** under the three initialization strategies (*random*, *jointEmb*, *XLM*); performances of optimizing **elbo** are also shown with XLM initialization. There are several observations to be highlighted here. **i)** Only the **standard** and the **elbo-dae** training protocols lead to decent performances, and the later requires using DAE loss as well and it is necessary to set the coefficient of the KL regularization term lower than 0.05. **ii)** Simply optimizing **elbo** leads to failure of training. **iii)** although **dae-only** seems to lead to very low performance through largely copying the input, if we continue training with the BT loss alone, we can surprisingly obtain similar (or sometimes even better) performance compared with **standard** training protocol (the +BT loss row), though the initial performance is very low (about 2 BLEU points), which indicating that it is not necessary that we have initial model with decent performance to make Co-Training successful. **iv)** For **bt-only**, if we continue with **standard** training, the final performance still struggles to reach that of **standard**; actually, for weak initialization methods (random, JointEmb), the model could even hardly learn copy, and stay failure all the time. This may imply that **bt-only** is learning *poisoned* inner representation. Actually, according to the information inequality (5), **bt-only** will make $\mathbb{I}(X, Z)$ to be very low, that is the output of the encoder can hardly identify the input sentence (Brunner et al., 2020), therefore data quality of Co-Training stays low all the time. We will design an experiment to prove this in Sec. 3.4.2.

---

[1] https://github.com/facebookresearch/XLM
[2] Please refer to the '*get-data-nmt.sh*' script in the XLM repo for more details; we use the default setting.

Table 3: Performances (Papineni et al., 2002) of the **standard** training and its two variants **dae-only** and **bt-only** under three initializations; together with **elbo**-*related* protocols with XLM initialization.

| Protocol | *random init.* | | | | *JointEmb init.* | | | | *XLM init.* | | | |
|---|---|---|---|---|---|---|---|---|---|---|---|---|
| | en-fr | fr-en | en-de | de-en | en-fr | fr-en | en-de | de-en | en-fr | fr-en | en-de | de-en |
| **standard** | 12.41 | 12.36 | 6.75 | 8.88 | 27.37 | 25.13 | 17.68 | 22.34 | 33.12 | 29.96 | 27.29 | 33.03 |
| **dae-only** | 1.39 | 1.35 | 2.68 | 2.75 | 2.20 | 2.19 | 2.74 | 2.79 | 2.32 | 2.31 | 3.40 | 9.61 |
| +BT loss | 11.19 | 10.71 | 7.79 | 9.66 | 26.75 | 24.96 | 18.93 | 23.31 | 33.14 | 29.88 | 26.67 | 32.87 |
| **bt-only** | 0.25 | 0. | 0. | 0.69 | 0.29 | 0. | 0.15 | 0.73 | 0.18 | 0.19 | 0.18 | 0.55 |
| +standard | 0.13 | 0. | 0. | 0.59 | 0.16 | 0. | 11.85 | 15.54 | 19.30 | 18.16 | 20.52 | 25.65 |
| **elbo** | - | - | - | - | - | - | - | - | 0. | 0. | 0. | 0. |
| **elbo-dae** | - | - | - | - | - | - | - | - | 32.67 | 29.81 | - | - |

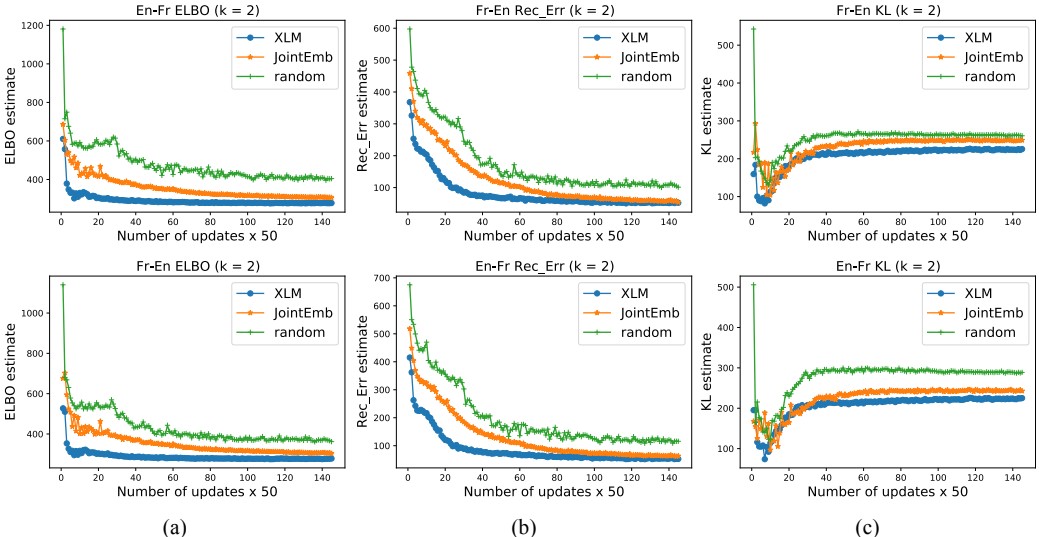

Figure 1: *Negative* ELBO learning curve visualization on WMT14 En-Fr validation set under **standard** training; each column represents (a) ELBO, (b) reconstruction error and (c) KL estimation.

## 3.2 VISUALIZING ELBO LEARNING CURVES

In this subsection, we set up to visualize the learning curves of ELBO, together with the two terms in ELBO, i.e. reconstruction error and KL divergence. The actual ELBO values are negative, but learning commonly means to minimize certain loss, so here we visualize the *negative* ELBO. The lower the value is, the better the ELBO is being optimized. We first demonstrate the learning curves of **standard** training, describe some observed phenomena, and then turn to visualize the curves of other failure training protocols. Note that, since we use two samples (k=2) in the Monte Carlo approximation, all the terms are *twice* the value as it should be.

Figure 1 (a) demonstrates the ELBO learning curves of the standard UNMT training under three initialization strategies. The overall ELBO on the two monolingual datasets is the sum of En⇒Fr and Fr⇒En directions. As you can see, across all initialization strategies, even though there is clear mismatch between the standard UNMT objective and the ELBO objective over the regularization term, we can conclude that **standard** training *implicitly* minimizes negative ELBO.

Figure 1 (b) is the visualization of the reconstruction error term within ELBO. It is self-evident that for **standard** training, the reconstruction loss represents the cross-lingual translation ability of the model, so in the original paper (Lample et al., 2018a) the reconstruction BLEU, which correlates well with the reconstruction error, is used for model selection without given bitext development set. Figure 1 (c) shows the visualization of the KL divergence term. It is interesting that for all initialization strategies, the KL value first goes down and then goes up quite a bit till convergence. This learning phenomenon can be summarized as: the **standard** training protocol tends to make the model first fit to behave like language model of the two languages and then fit to translation model

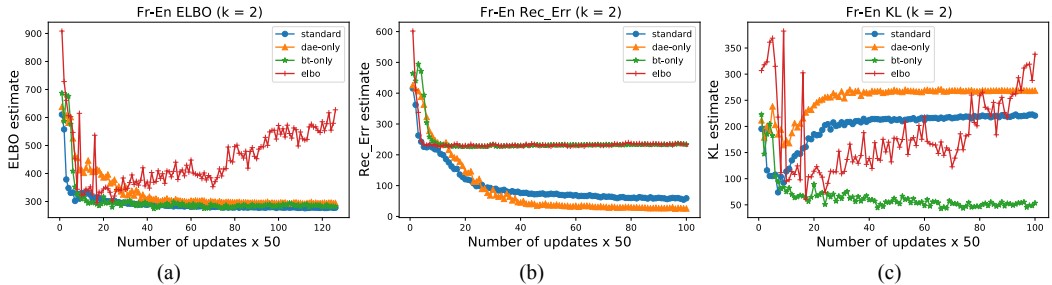

Figure 2: *Negative* ELBO learning curve visualization on WMT14 En-Fr validation set for all failure variants comparing with the **standard** training protocol with XLM initialization.

in later stage. And the 'going-up' reflects the existence of large distance between the distribution of a language model $p(y)$ and the translation model $p(y|x)$, given same target $y$.

Next, we continue to visualize the ELBO curves of some failure variants of the training protocol. In Figure 2, we draw all the ELBO learning curves for those training protocols presented in Table 2 that fail to train a well-performed model under XLM initialization. As you can see in Figure 2 (a), other than **standard**, before (30x50=) 1500 updates, all variants seem to reach similar ELBO values. As the training goes on, **dae-only** and **bt-only** tend to have exactly the same ELBO value with the **standard** training protocol. However, the reconstruction error of **bt-only** is much higher than **standard**, while the KL divergence of **dae-only** is much higher than that of **standard** as well. Both a low reconstruction error and a low KL distance are necessary for successful training. For **elbo-only**, there is a quick *posterior collapse* phenomenon at initial training (before 1k updates) (in Figure 2 (c) KL becomes very low), however, then the KL slowly goes up, which might be resulted from the unstableness of ELBO optimization with REINFORCE (He et al., 2020). This indicates that only requiring ELBO to be optimized *as a whole* is only the *necessary* but *not* the *sufficient* condition of successful learning the target translation task.

### 3.3 WHY MINIMIZED ELBO CAN *still* LEAD TO TRAINING FAILURE?

An intuitive explanation is that unsupervised learning through marginal likelihood maximization is *under-determined*. There are many plausible tasks like language modeling, paraphrasing, simple sequence copying, translation, that satisfy the inductive bias of the parameterization, and freely learning with objectives like ELBO can make the model learn any of the plausible tasks if optimization finally converges. And learning any one of them can induce minimized ELBO. So what tasks on earth have **dae-only** and **bt-only** finally learned respectively? Table 7 in appendix demonstrates the decoding behavior of the final models given certain source inputs. We can assume that **dae-only** degenerates to the sequence *copy* task while **bt-only** degenerates to the *language modeling* task.

#### 3.3.1 ANALYSIS ON FAILURE OF DAE-ONLY

**Learned *copy*** Figure 2 (b) informs us that **dae-only** has even lower reconstruction error than **standard**, which means that even if **dae-only** is not trained with the language embedding feeding of another language in the target-side, it can still minimize the reconstruction loss when feed with the target language embedding. However, Table 7 demonstrates that **dae-only** has learned almost perfect sequence *copy*. Here we use the definition of cross-lingual RecErr to clarify this phenomenon, since unconstrained decoding might generate tokens that come from source language and this might prevent us from distinguishing RecErr of *mono-lingual* reconstruction (copying, paraphrasing) and *cross-lingual* reconstruction (translation).

In previous subsection, we draw all the learning curves in Figure 2 (b) based on naive RecErr without considering the above situation. Here in Figure 3, according to a modified Definition A.2, we can draw the *cross-lingual* RecErr for **standard**, **dae-only** and **bt-only** accordingly. As shown, the cross-lingual RecErr curve is much higher than the RecErr for **dae-only**, and it is the highest among all, indicating that, essentially, the target translation task is learned *only* by learning a *low* cross-lingual reconstruction loss. That is why later work like Liu et al. (2020) directly uses constrained

Table 4: Token-to-token mutual information estimated over the generated pseudo-bitext.

Table 5: Controlling the update frequency ratio of DAE and BT losses under XLM init.

|  | *greedy* | | *sample* | |
|---|---|---|---|---|
|  | en-fr | fr-en | en-fr | fr-en |
| **standard** | 0.55 | 0.53 | 0.46 | 0.52 |
| **bt-only** | 0.06 | 0.08 | 0.11 | 0.12 |
| **dae-only** | 0.57 | 0.50 | 0.56 | 0.58 |
| **dae-only**(cross) | 0.60 | 0.49 | 0.43 | 0.37 |
| **random** | 0.27 | 0.27 | 0.27 | 0.27 |

| $\tau$ | en-fr | fr-en |
|---|---|---|
| 0.02 | 24.37 | 22.82 |
| 0.05 | 27.21 | 24.55 |
| 0.1 | **33.46** | 29.78 |
| 1 | 33.12 | 29.96 |
| 2 | 32.02 | 29.18 |
| 5 | 2.30 | 2.29 |

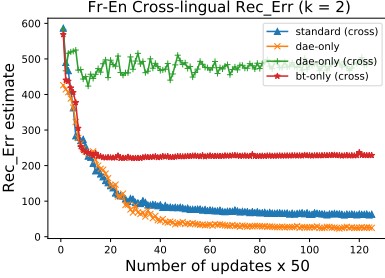

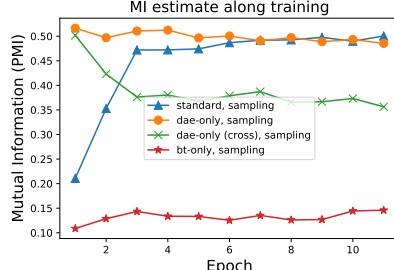

Figure 3: The learning curve of cross-lingual RecErr based on Definition A.2; the RecErr of **dae-only** is shown for comparison as well.

Figure 4: The mutual information estimation of the pseudo-bitext generated by *sampling*-based decoding along the training process.

decoding for BT to accelerate training. We also calculate the correlation between RecErr/cross-lingual RecErr and BLEU in Figure 5 (see the appendix), the later is much better correlated (-0.37 verses -0.87) with final performance. Moreover, in Table 4, we calculate the mutual information in pseudo-bitext generated by **dae-only** under constrained decoding, suprisingly we find that the MI is very high. However, since **dae-only** does not expose the model on such data, it never learns the cross-lingual alignment in the data, which indicates the important role of BT that can actually learn such cross-lingual MI. These findings support our practical sufficient condition: a lower cross-lingual RecErr is more likely to make UNMT training successful.

### 3.3.2 ANALYSIS ON FAILURE OF BT-ONLY

**Degeneration to LM** As you can see in Figure 2 (c), the sentence-level KL distance between **bt-only** and the language model is very small, much lower than **standard** and others. This indicates that the learned behavior of **bt-only** may resemble the behavior of a language model. That is, the UNMT model learned with **bt-only** might largely ignore the potential predictive information contained at the source side, and only relies on decoder's LM prior.

**MI of Pseudo-Bitext** The reason why such degeneration happens during the training process can be intuitively visualized by showing the mutual information contained in the pseudo-bitext generated with online iterative BT. Table 4 shows the mutual information of the final checkpoints obtained by **standard**, **bt-only** and **dae-only**. As you can see **bt-only** has the lowest mutual information between source and target of the generated bitext. And even if we instead use sampling for generation, the mutual information is still lower than random bitext ($0.12 < 0.27$). We also conduct experiment when at BT phase the model uses sampling instead of greedy, this will alleviate the degeneration a little bit, but the learning still fails. We further draw the mutual information of pseudo-bitext along training in Figure 4, and **bt-only** stays low all the time, while **standard** have growing values.

### 3.4 THE CRITICAL ROLE OF BT AND DAE LOSSES

#### 3.4.1 BT LOSS IS THE MAIN TASK WHILE DAE LOSS THE AUXILIARY

In the standard UNMT training protocol, we can think of consecutive learning of DAE and BT losses as multitasking (Caruana, 1998). However, since the (cross-lingual) BT-loss is directly related to

Table 6: The estimated mutual information between encoder input $X$ and output $Z = enc(X)$.

| protocol | standard | bt-only | elbo-only | XLM init. |
|---|---|---|---|---|
| en | 7.603 - 0.079 | 7.603 - 2.863 | 7.603 - 4.299 | 7.603 - 0.142 |
| fr | 7.449 - 0.096 | 7.449 - 2.571 | 7.449 - 3.980 | 7.449 - 0.123 |

$I(x, \hat{y})$ which is used to guarantee the success of translation in principle as mentioned in Sec. 2.3. we may think BT task as the *main* task and DAE task as the *auxiliary* task with respect to the target translation task. This section gives an affirmative answer to this intuition. Since DAE and BT losses are updated one after another, we can control the update frequency ratio $\tau$ between them. This can control the degree of the model learning towards certain loss and $\tau = 1$ corresponds to the standard setting. Table 5 shows the results. The larger $\tau$ is, the more frequent DAE loss is being updated. As you can see, it seems the more DAE loss is updated, the training tends to become the **dae-only** setting, that is, the model starts to learn copy instead of translation. On the contrary, it does not hurt so much even BT is updated 10 times more than DAE ($\tau = 0.1$), though increasing BT updates a lot (larger than 20 times, $\tau = 0.05$) will definitely degrade final performance, which reflects the important role of DAE's regularization effect on BT, that is to constrain online inference at BT phase so as to prevent the model from learning unexpected noise.

### 3.4.2 DAE NOT JUST HELPS WITH LANGUAGE MODELING, BUT PRESERVES MI

In previous works DAE loss have been recognized as learning word order aka. the language modeling ability of the UNMT model Lample et al. (2018a); Artetxe et al. (2018); Lample et al. (2018b); Yang et al. (2018); Kim et al. (2020). Here we would like to clarify that the most critical functionality of DAE is not just learning language model, but at least *preserving the MI* between encoder input and output which matches the *necessary condition* we introduced in Sec.2.3 2.3. As a result, it can prevent the model from degeneration during online BT. To this end, we first experiment with a different version of DAE loss that ignores word order, that is, when constructing DAE loss from $x$ we first **permute the order** of $x$, denoted by Perm$[x]$ and then optimize $-\log P_{s \to s}(\text{Perm}[x]|C(\text{Perm}[x]))$ instead. For XLM initialization, the final performance only drops from 33.12 to 31.02. This indicates that DAE are not only learning word order, but something more critical, i.e. preserving the MI between input $X$ and encoder output $Z$.

We verify this by estimating the MI between $X$ and $Z$ for encoders trained with **standard**, **bt-only**, **elbo-only**, and a baseline encoder initialized from XLM. Then we only train a random initialized decoder over the encoder. In Table 6, each entry consists of two terms with the first term an estimation of $\mathbb{H}(X)$ and the second term an estimation of $\mathbb{H}(X|Z)$. As you can see, without the regularization effect of DAE, **bt-only** and **elbo-only** has very large entropy of $X|Z$, even much larger than the XLM initialized encoder. This can explain the previous highlighted phenomenon in Table 3, that is, after **bt-only**, if we continue train with **standard**, only XLM initialization can recover certain performance while the other two stay failure. The reason is that without DAE, the encoder representation is being contaminated without contain any useful information of X. Actually, DAE not only preserve the MI between encoder input and output, in Table 4 and Figure 4, we have drawn the cross-lingual MI, i.e. $\mathbb{I}(X, Y')$, contained in pseudo-bitext generated by **dae-only** trained model, it seems that **dae-only** with XLM initialization have already learned initial word-to-word translation ability. This can be further leveraged by online BT to learn towards real sentence-by-sentence translation.

## 4 CONCLUSION

This paper conducts thorough *comparative studies* of the **standard** UNMT training protocol and its variants from two theoretical views, i) marginal likelihood maximization and ii) mutual information. We find that **standard** training implicitly optimizes ELBO so as other failed variants, indicating the importance of DAE as a *regularization* for helping the model learn the correct target task. Low BT losses (*cross-lingual* reconstruction loss) is a self-evident *sufficient* condition for successful training of UNMT, and the high mutual information between $X$ and $Z = enc(X)$ is a *necessary* condition for preventing the model from degeneration. In addition, DAE loss plays the role of preserving $\mathbb{I}(X, Z)$ as well as $\mathbb{I}(X, Y')$; meanwhile, online BT is the *main* task that enables the model to actually learn from emerging *cross-lingual* signals unveiled by DAE in the pseudo-bitext.

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

# A  APPENDIX

## A.1  RELATED LITERATURE

### A.1.1  BETTER INITIALIZATION STRATEGIES FOR UNMT

In our introduction, we have mentioned that if the encoder-decoder model has already achieved certain decent initial performance, then by using BT loss solely can reach comparative or even better performance than XLM initialization. MASS (Song et al., 2019) is the first work that achieves this, they pretrain a sequence-to-sequence model for predicting a span of a sentence given the span-dropped sentence as input. They find in their experiments that dropping half of the sentence as a contiguous span achieves the best result. Recently, mBART (Liu et al., 2020) extends the idea of denoising pre-training on Lewis et al. (2020a) to the multilingual setting. They pre-train sequence-to-sequence model on monolingual corpora of 25 languages, and only use BT loss to finetune the model for UNMT. In their paper, they claim that when relying only on BT, they use constrained decoding to obtain sentence on the other language at initial epochs to overcome the *copy* issue. Conceptually, mBART actually redefines the role of DAE loss as a pre-training objective, and this largely matches our findings in Table 3's third row (+BT loss), that is we use DAE loss alone to train the model under certain initialization and then continue to train it solely with BT, and this matches the **standard** training protocol. However, when using DAE as pre-training loss from random initialization, the model could only achieve 10+ BLEU far less than 30+. Some reason for this gap might be: 1) the noisy function $C$ of the DAE loss in **standard** training is a little bit different than that in BART; 2) we haven't use such larger corpora for pre-training. We think to find out the reason might have significant contribution to the community on questions like: a) *what is the data-scale for pre-training to actually work?* and b) *what kind of self-supervision is more effective than others?* Other works like Ren et al. (2019) directly conduct masked language model pre-training with explicitly constructed cross-lingual prediction signals, which is obtained from cross-lingual word translation techniques (Conneau et al., 2017). Although they do not apply their method on sequence-to-sequence pre-training, and their method could be applied directly to MASS.

### A.1.2  PRACTICAL ISSUES OF UNMT

Recently, several works start to criticize the practicality of the standard UNMT training. Kim et al. (2020) and Marchisio et al. (2020) both claim that domain mismatch of the two monolingual corpora and the dissimilarity of the language pair correlate well with performance degradation. In Kim et al. (2020), they investigate pratical scenario with three factors: i) linguistic distance; ii) availability of large-scale bitext; iii) availability of large-scale monolingual text. They instantiate the factors with 5 chosen language pairs and find that the standard UNMT training protocol only works for pairs with close linguistic distance and abundent monolingual text. Similar to Kim et al. (2020), Marchisio et al. (2020) also conduct an extensive empirical evaluation of for unsupervised machine translation using dissimilar language pairs, domains and authentic low-resource languages. However, instead of using pure NMT model, they also rely on statistical machine translation model for warming up the NMT model, which is not the standard training protocol that we have investigated. In fact, although using different training protocol, they find similar observations as that in Kim et al. (2020).

### A.1.3  IMPROVED TRAINING PROTOCOL

Tran et al. (2020) proposes a novel cross-lingual retrieval method for finding comparative sentence pairs from monolingual corpora of the two language. They use the multilingual pre-trained encoder of mBART (Liu et al., 2020) to get universal semantic representations of sentences (by doing this, they just average the token-level vectors from mBART as a single vector for nearest neighbor search) for retrieval of potentially aligned sentence pairs for iterative self-supervised training. This method resembles that of Wu et al. (2019) who use the UNMT model's own encoder representation instead of a self-supervised pre-trained encoder, and can be seen as its multilingual pre-training extension. All of the above proposed specific training methods for UNMT together with recent paraphrase-based pre-training objective (Lewis et al., 2020b) can all be thought of as implicit maximum likelihood training (Li & Malik, 2018), since the retrieval phase is certain instantiation of k-Nearest-Neighbor search. Duan et al. (2020) also proposes a new training method by constructing mixed code pseudo-

bitext. Their method proves the effectiveness of using unsupervisedly induced bilingual lexicon as *'anchor'* for better preventing BT from learning self-generated noised bitext.

## A.2   THE STANDARD TRAINING PROTOCOL

Please refer to Algorithm 1 for a detailed description of the standard training protocol.

---

**Algorithm 1:** The Standard UNMT Training Protocol

**Input:**
 A large-scale pre-training corpus $\mathcal{D}_{pt}$;
 Two monolingual fine-tuning corpora $\mathcal{D}_s$ and $\mathcal{D}_t$;
 An untrained encoder-decoder $\mathcal{M}_{\theta_{e,d}}$ with $\theta_e$ and $\theta_d$
 and specifically $\theta_e^e \subset \theta_e, \theta_d^e \subset \theta_d$ as the embeddings.

**Output:**
 The estimated UNMT model $\mathcal{M}_{\theta_{e,d}}$.

 1: *// initialization*
 2: Learn joint BPE code on $\mathcal{D}_s \cup \mathcal{D}_t$;
 3: Apply BPE to the pre-training corpus $\mathcal{D}_{pt}$;
 4: **if** *pretrain* = 'JointEmb' **then**
 5:     Apply fastText on $\mathcal{D}_{pt}$ to learn embeddings
 6:     Initialize $\theta_e^e$ and $\theta_d^e$ with the learned joint embeddings
 7: **else if** *pretrain* = 'XLM' **then**
 8:     Train $\mathcal{M}_{\theta_{e,d}}$ with self-supervised losse(s) on $\mathcal{D}_{pt}$;
 9:     Initialize the $\theta_{e,d}$ with the learned parameters;
10: **else**
11:     Initialize the $\theta_e$ and $\theta_d$ randomly;
12: **end if**
13: *// fine-tuning*
14: $step = 0$;
15: Sample monolingual batch $b_s \in \mathcal{D}_s, b_t \in \mathcal{D}_t$
16: Construct denoising language modelling loss
     according to Eq. 1;
17: *Update model parameters using ADAM by back-propagating Eq. 1*;
18: Sample monolingual batch $b_s \in \mathcal{D}_s, b_t \in \mathcal{D}_t$;
19: Use $\mathcal{M}_{\theta_{e,d}}$ to translate each batch to the other
     language side as $\hat{b}_t$ and $\hat{b}_s$;
20: Construct back-translation loss according to
     Eq. 2 on the two paired bilingual
     batches $(\hat{b}_t, b_s), (\hat{b}_s, b_t)$;
21: *Update model parameters using ADAM by back-propagating Eq. 2*;
22: $step \mathrel{+}= 1$;
23: **if** $step$ = MAX_STEP **then**
24:     End training;
25: **else**
26:     Go to line 15;
27: **end if**
28: **return** $\mathcal{M}_{\theta_{e,d}}$;

---

## A.3   CONSTRAINED DECODING AND CROSS-LINGUAL RECERR

Let us take En-Fr translation task as an example. Since English language and French language share a large amount of vocabularies, and the sharing will be enhanced due to subword tokenization, i.e. BPE. The percentage of shared vocabulary of En and Fr are above 70% in our data setting. Thus, here we leverage a simple but effective heuristic to divide the shared vocabulary into En-*dominant* and Fr-*dominant* subones. The idea is to use a token's frequency ratio over the English monolingual and French monolingual corpus as an *indicator* of language it mostly likely belongs to. Say given a token $t$, we can compute its frequency ratio as $r = freq_{en}(t)/freq_{fr}(t)$. If the ratio $r$ is larger than certain threshold $\tau$, we say $t$ belongs to English since it is mostly in use in English than in French, and vice versa. In experiment, we set $\tau = 2$ to get reasonable vocabulary division. And then during decoding, we set the logits of tokens in the other language to $-\infty$ for satisfying our constraint.

Table 7: Some examples of the degeneration phenomenon for **dae-only** and **bt-only** variants.

| | **dae-only** fr-en |
|---|---|
| src | L' avocat de Manning a déposé une plainte formelle pour les traitements subis par Manning en janvier 2011 . |
| prd | L' avocat de Manning a déposé une plainte formelle pour les mauvais traitements subis par Manning en janvier 2011 . |
| tgt | Manning 's lawyer filed a formal objection to Manning 's treatment in January 2011 . |
| | **bt-only** fr-en |
| src | Le télescope Hubble a observé la naissance de ces étoiles dans la galaxie spirale M83 . |
| prd | The is the only way to ensure that the European Union is able to act . |
| tgt | View of the European Southern Observatory ( ESO ) in the Chilean Andes . |
| src | Le télescope Hubble a observé la naissance de ces étoiles dans la galaxie spirale M83 . |
| prd | The is the only way to ensure that the European Union is able to play its part in the fight against terrorism . |
| tgt | This star birth was captured by the Hubble telescope in the M83 spiral galaxy . |

**Definition A.1. (RecErr)** Given an UNMT model $\mathcal{M}$, its reconstruction error on a mono-lingual text $\mathcal{D}$ is defined as: $\frac{1}{|\mathcal{D}|} \sum_{x \in \mathcal{D}} \frac{1}{|x|} \log P(x|\mathcal{M}[x])$. Here $\mathcal{M}[x]$ denotes the model's output sequence through greedy decoding or sampling.

**Definition A.2. (Cross-Lingual RecErr)** Given an UNMT model $\mathcal{M}$, the cross-lingual reconstruction error on a mono-lingual text $\mathcal{D}$ is defined as: $\frac{1}{|\mathcal{D}|} \sum_{x \in \mathcal{D}} \frac{1}{|x|} \log P(x|\mathcal{M}^c[x])$. Here $\mathcal{M}^c[x]$ denotes the result of *constrained decoding* for predicting a sequence of tokens in the other language.

## A.4 COMPUTE ELBO

Since in our following experiments, we are going to visualize the learning curve of ELBO along the training life cycle, we should be able to empirically compute the two terms of ELBO which both involve expectation over $q_\phi(y|x)$. Instead of using greedy decoding for obtaining samples, we use sampling (k=2) to compute reconstruction error and KL divergence, and both terms are computed via Monte Carlo method. For the KL term, $p_\theta(y)$ is a language model trained on $D_t$. Instead of using token-level ELBO like that used in He et al. (2016), we do not normalize the ELBO values by the number of tokens in $y$, and use the sentence-level ELBO value for visualization.

## A.5 ESTIMATING $\mathbb{H}(X)$, $\mathbb{H}(\mathbb{X}|\mathbb{Z})$ AND $\mathbb{I}(X, Y')$

For estimating $\mathbb{I}(X, Z)$ which is the MI between discrete and continuous *r.v.*s, we use the equality $\mathbb{I}(X, Z) = \mathbb{H}(X) - \mathbb{H}(X|Z)$, and then estimate $\mathbb{H}(X)$ and $\mathbb{H}(X|Z)$ respectively. For the entropy of $X$, we use a 1-gram language model on the same training corpus for training the UNMT model, and use the average *token-level* entropy as a surrogate; as for the entropy of $X|Z$, we train an extra *reconstruction model* over a fixed UNMT encoder to make sure we are using the representation from certain UNMT model, then we use the reconstruction model (a decoder)'s token-level entropy as a surrogate. This is motivated by Gao & Chaudhari (2020) who leverage the reconstruction error as a measure for how much information has discarded in the hidden representation, and also similar to recent probing methodology (Conneau et al., 2018).

For estimating $\mathbb{I}(X, Y')$, we use the *token-by-token* point-wise mutual information (PMI) over some pseudo bitext as a surrogate to the *sentence-by-sentence* MI. Note that since we use different estimators (continuous v.s. discrete) for computing $\mathbb{I}(X, Y')$ and $\mathbb{I}(X, Z)$, moreover, according to Pimentel et al. (2020), the model-based estimation of $\mathbb{H}(X) - \mathbb{H}(X|Z)$ is a lower bound, so it is hard to compare between $\mathbb{I}(X, Y')$ and $\mathbb{I}(X, Z)$. But values within one estimator are comparable.

Here we give a detailed introduction of how we estimate the above statistics. For estimating the first two terms, we follow the formula introduced in Pimentel et al. (2020) to estimate the entropy:

$$\mathbb{H}_{q_\theta}(X; C) \approx -\frac{1}{N} \sum_{i=1}^{N} \log q_\theta(X|C). \tag{6}$$

Here if $C = \varnothing$ is null, we use it to estimate $\mathbb{H}(X)$; or if $C = Z$, we use it to estimate $\mathbb{H}(X|Z)$.

**Estimate** $\mathbb{H}(X)$ We estimate the *token-level* entropy instead of sentence-level, that is, $X$ denotes a token *r.v.*. The $q_\theta$ we use is an 1-gram language model on the concatenated En, Fr corpora.

**Estimate** $\mathbb{H}(X|Z)$ The $q_\theta$ we use to estimate $\mathbb{H}(X|Z)$ is a Transformer decoder over fixed encoder that provides hidden representations $Z$. So we first train a decoder over the training corpus, and then use the decoder to provide the $\log$ likelihood of every token $X$.

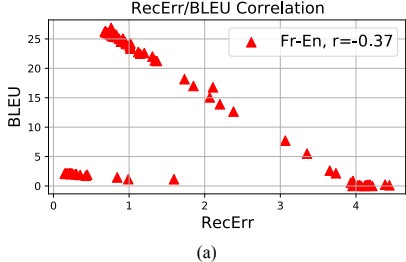 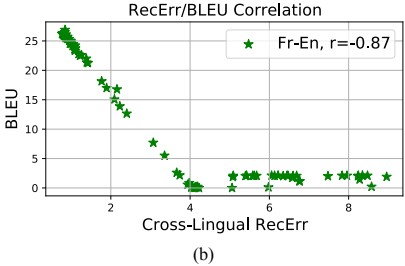

(a)                (b)

Figure 5: Correlation of the mono-/cross-lingual rec_err and the final performance.

Here, to make $\mathbb{H}(X)$ and $\mathbb{H}(X|Z)$ comparable, the estimates of $\mathbb{H}(X)$ and $\mathbb{H}(X|Z)$ are calculated on the **same** training set held-out, about 50k sentences.

**Estimate** $\mathbb{I}(X, Y')$ Given a large amount of pseudo-bitext, we use the point-wise mutual information as token-level estimates of the actual mutual information in paired sentences, that is:

$$\text{PMI}(X, Y') = \frac{1}{N} * \frac{1}{l_x \cdot l_y} \sum_{x_i, y_j} \frac{P(x_i, y_j)}{P(x_i)P(y_j)}. \tag{7}$$

