# OpenReview forum: "Demystifying Learning of Unsupervised Neural Machine Translation"
_ICLR.cc/2021/Conference — Reject_

### Official Review · AnonReviewer1 · 2020-10-27
**Interesting observations but unclear takeaways and confusing writing**

**Rating:** 5
**Confidence:** 3

**Review:**

This paper takes a closer look at the inner workings of unsupervised MT training.

The authors provide two alternate views on the backtranslation+DAE objectives used in unsupervised-MT. This interpretation sheds more light on the relationship between the two: for example, it appears that the DAE loss is critical to preserve mutual information between input and encodings, thus preventing the translation models from degenerating into unconditional language models. The authors' conclusions are well supported by experiments. However, I find that the paper falls short by being purely descriptive. While the many observations made by the authors are interesting, the reader is left hanging trying to figure out what to do with them. What is the point of these observations? Why are they useful? This paper is missing a second, more "prescriptive" half answering these questions.

**Pros**

- Sheds light on the role of the two loss terms
- Convincing experiments: the authors claims are well supported by their experiments

**Cons**

- Purely descriptive: what is the takeaway? How can we improve unsupervised MT based on these findings? Better model selection? No need for denoising if the autoencoder is trained less?
- The paper is very hard to read. I attribute this to the abundance of notations and acronyms, as well as the unwieldy phrasing. In addition, the progression of the experimental section is hard to follow. This could be remedied by summarizing and clarifying the contributions better. Right now, it reads like a list of facts about UNMT (this is also related to my first point)
- I am not 100% convinced that the mutual information between model prediction and input is a good measure of MT success. Wouldn't that make a model that just copies its input successful (if H(x)>c )? Could the authors provide more intuition on why that is a good metric?

**Remarks**
- Grammar errors and typos: there are a lot of grammar errors and typos. I understand that the authors may not be native speakers of English. Nevertheless, the amount of errors is high enough to be distracting. I highly encourage the authors to seek proof-reading. Some examples:
   * "why this standard training protocol is ~possible to be~ successful"
   * "some fundamental questions are still remained unknown" -> "some fundamental questions remain unanswered"
   *"~If~ We denote Y 0 = M(X) as the random variable (r.v.) generated by the model M over X."
   * "repoitory" -> "repository"
   * "BT LOSS IS THE MAIN TASK WHILE DAE LOSS THE AUXILIARY" -> "BT IS THE MAIN TASK WHILE DAE IS AUXILIARY"
- Style: I understand that this is a personal preference to some extent (which is why I didn't count it as a con), but the writing style is not very "academic": Examples: addressing the reader in the second person (2.2: "As you can see"), awkward/informal phrasing (1: "we present a formal definition on what does it mean to [...], "3: "As promised in Section [...]")
- There are vague statements that should either be made more precise or removed altogether. For example in Sec. 2.3: "Therefore, if (Y 0;X) gradually contains more and more bilingual knowledge," what does "containing more bilingual knowledge" mean for a pair of sentences?
- I didn't quite follow the point of the fine-tuning experiments in 3.1 (the +bt loss and +standard rows in table 3). Specifically, I'm not sure I understand what the authors mean by saying that this shows that bt-only "poisons" the model. What is meant by "poisoning"? I'm not sure how this fits in the overall narrative of the paper, and I found this observation confusing more than anything else.

---

Post rebuttal:

I thank the authors for responding to my questions. While some minor points are cleared up, I am still not completely satisfied with the rather vague notion of "bilingual knowledge" (what even is the "correct axiomatic translation correspondence", is it the (idealistic) true data distribution one tries to model?). Similarly, the use of the MI as a measure of success is still unsatisfying to me since it relies on a "trick" (mismatch of vocabulary between source and target) which might not even be relevant in practice (since most models use sub-words anyway).

Overall, I think this research direction is promising, but I keep my recommendation the same. The paper would greatly benefit from another round of revision to clear up these points and clarify the presentation if it is to be useful to the research community.

---

> ### Author Response · Authors · 2020-11-23
> **We think the theoretically-sounded explanation should go first, and we will discuss some practical takeaways in the updated version.**
>
> Thanks a lot for your detailed comments. Specifically, we really appreciate you concern on our poor English writing, we will definitely ask native speakers to check our wordings! Furthermore, we would like to clarify some points here.
>
> Q1. *ELBO-only protocol fails to train*
> - I think this is exactly the information the analogy tries to emphasize, that is, optimizing ELBO is not sufficient to lead to successful training of the desired task, since there are hundreds of tasks that optimize ELBO implicitly as well. What is more important is to design regularization that resembles target task in certain way so as to bias the optimization towards finding the targeted solution.
>
> Q2. *gradually contains more and more bilingual knowledge*
> - Actually, we mean **bilingual knowledge** as the correct axiomatic translation correspondence (for example word-to-word translation). Since at the beginning phase of BT training, the word-to-word bilingual signals is weak and gradually grows as training proceeds under standard training protocol.
>
> Q3. *I didn't quite follow the point of the fine-tuning experiments in 3.1*
> - Firstly, the row 2 and row 3 (dae-only, +BT loss) are trying to convey that dae-only can act as pre-training for BT-only training. Though dae-only might degenerate to nearly perfect copy of the source sentence, it actually plays the role of **pre-conditioning** for BT-only training. This actually achieves comparable performance with the standard training.
> - Secondly, the row 4 and row 5 (bt-only, standard) are trying to convey that if we use bt-only to train the model and it degenerates to a target-side language model that completely ignores source information. If we go on training with the standard training (which will succeed with the original 3 initialization strategies) but fail to train the model to similar performance as before. For example for XLM init. standard training reaches 32.67 for en-fr, but after  XLM init. + bt-only training as the new initialization, standard training could only reaches 19.30 far less than 32.67. This indicates that bt-only has made the XLM init. far less effective. So we say that bt-only poisons the model's initialization.
>
> Q4. *I am not 100\% convinced that the mutual information between model prediction and input is a good measure of MT success*
> - We should emphasize that the mutual information is computed between two $X$ and $Y'$ where they are composed of tokens from the two languages separately so copy is not allowed. Based on this, the cross-lingual mutual information is a very simple model selection strategy when no bilingual development set is given.

---

### Official Review · AnonReviewer3 · 2020-10-28

**Rating:** 6
**Confidence:** 3

**Review:**

This paper aims to explain the learning of unsupervised neural machine translation(UNMT) theoretically from two perspectives:
1)` marginal likelihood maximization, where the objective of UNMT is analogous to ELBO and it's two terms are visualized during the training of standard UNMT.
2) information-theoretic view, where the authors derive the success of UNMT training as maximizing the mutual information between source and target translation (sufficient condition), as well as source input and its encoder output (necessary condition). Later in the experiments, the paper presents the MI during training to support their arguments.

Pros
1. This paper attempts to explain UNMT theoretically, which is a significant step for understanding UNMT inspiring the future direction of UNMT.
2. The experiments are properly aligned with the theoretical arguments made in the paper and give good empirical explanations.


Cons:
1. It seems that we can't really evaluate if a model meets the real sufficient condition using the practical sufficient condition.
2. Although this paper provides propositions that what is a success UNMT training, it could've been better if showed how we can create a better UNMT system based on these definitions.
3. ELBO-only protocol fails to train, which makes the analogy b/w *standard* and ELBO in the perspective of marginal likelihood maximization not pretty much hold thus less convincing.

----

Questions:
1. fig 4: I don't really understand why dae-only always preserves a high(est) MI during training. The authors barely explained anything in terms of this.
2. though c in the def 2.1 & the sufficient condition is a conceptual quantity, to what extend we can consider a UNMT model is successfully learned? Even an intuitive thought is welcome.


------
Reason for score: This paper gives a theoretical view of understanding UNMT, which is a good contribution. Some of the analogy and propositions are not well supported by the empirical results. This paper could have given some future directions based on their theoretical understanding.

---

> ### Author Response · Authors · 2020-11-23
> **We hope that some findings can point to new research directions such as emergence of multilinguality from unaligned training, e.g. pre-training with LM objectives.**
>
> We really thank you for your positive comments as well as your critics!
> We would like to response to some of your points in your ***Cons*** comments as well as to your two questions.
>
> Q1. *fig 4: I don't really understand why dae-only always preserves a high(est) MI during training.*
> - Sorry for making you confused to understand here since our explanations are scattered across different subsections. We will reorganize some of the statements and descriptions to make the points clearer.
> - Let's give an explanation to your question here. The most informative curves of this figure are the **dae-only** and **dae-only (cross)** curves. **dae-only**, as shown in the first example in Table 7, degenerates to learning nearly perfect **copy** of the source, this will make the mutual information the highest since $Y'$ is almost $X$, i.e. $Y'=X$. However, if we use **constrained decoding** to force the model predict tokens **only from** target vocabulary instead of the source, that is, the model generates as few copied tokens in $Y'$ as possible. However, the MI is still very high during training, which means that DAE-only learns more than copy but some initial bilingual knowledge like word-to-word translation.
>
> Q2. *to what extent we can consider a UNMT model is succesfully learned.*
> - This is a very difficult question to answer in theory, and it also relates to your first critic in the `Cons' part.
> We think one possible definition of success is to let the end user or researcher to set up the value of the cross-lingual mutual information $c$. However, we use Propotion 1 as simply a conceptual framework for deriving the necessary and sufficient conditions for clear interpretation. For us, we think a BLEU score of nearly 10 could be treated as successful training, since in some low-resource translation tasks for example some language pairs in the IWSLT speech translation task, the BLEU score is about 10. Actually by inspecting the predictions, the model is hallucinating a lot, but also capture some main semantics from the source.
>
> Q3. *ELBO-only protocol fails to train*
> - I think this is exactly the information the analogy tries to emphasize, that is, optimizing ELBO is not sufficient to lead to successful training of the desired task, since there are hundreds of tasks that optimize ELBO implicitly as well. What is more important is to design regularization that resembles target task in certain way so as to bias the optimization towards finding the targeted solution.

---

### Official Review · AnonReviewer4 · 2020-10-29
**Interesting, but sometimes unclear, discussion of unsupervised neural machine translation**

**Rating:** 4
**Confidence:** 3

**Review:**

This paper attempts to explain why UNMT using back-translation (BT) and denoising autoencoding (DAE) has been successful.

I generally found this paper difficult to follow. Both the language and logic of its arguments were often not very clear to me.

Section 2 presents some theoretical arguments, or analogies.

Section 2.2 draws an analogy between UNMT and ELBO; the reconstruction-error terms are similar but the regularization terms are different.

Section 2.3 tries to connect UNMT to mutual information, but I didn’t understand why Definition 2.1 tries to define the “success” of UNMT in terms of MI (wouldn’t setting M to the identity function achieve maximal MI?).

Both of these analogies are very rough, and I feel that this section falls short of providing understanding of UNMT.

Section 3 presents some experiments to support the arguments made in Section 2.

Section 3.2 shows that UNMT minimizes the ELBO term that is similar to a term in the UNMT loss and doesn’t minimize the ELBO term that isn’t similar to a term in the UNMT loss.

Section 3.3 argues that using only the BT loss will cause UNMT to degenerate to generating fluent target-language sentences, while using only the DAE loss will cause UNMT to degenerate to copying the source to the target.

Section 3.4.1 argues that BT is more important than DAE because UNMT performs best when BT is weighted 10 times more heavily than DAE.

Section 3.4.2 argues that DAE’s job is to ensure that the encoder does not lose information.

Overall, it seems to me that these findings agree with the intuitive understanding of UNMT that was already present in the UNMT literature, with the exception of the finding in 3.4.1 that the BT term should be given higher weight than the DAE term.

While I think that explaining why UNMT works is an excellent research goal and there are some interesting ideas here, I do not think that this paper is ready for publication yet.

---

> ### Author Response · Authors · 2020-11-23
> **We modestly dub our work as the first attempt to ground the success of UNMT training with theoretical insights.**
>
> We sincerely thanks you for your critics and comments you listed according to each section of our paper. Here, we would like to response to some of your most concerned arguments.
>
> Q1. *wouldn't setting M to the identity function achieve maximal MI?*
> - Yes, the identity function is can have its inverse so preserve every information in X thus $\mathcal{I}(X, Y'=X)$ the maximal. However, since here we actually consider the translation task where $Y'$ should be a sentence composed by words from the target vocabulary. So the mutual information is actually cross-lingual.
>
> Q2. *but I didn't understand why Definition 2.1 tries to define the success of UNMT in terms of MI*
> - In practice it is hard to say the UNMT training is successful. Actually by saying UNMT training is successful we mean that the training do not degenerate and the model can at least achieve BLEU say about 10 BLEU points which means the model has learned to translation some word correctly.
> - In theory, according the the above practical consideration, the higher the mutual information between two sentences of two languages, the better the translation quality is. So we use a user-defined constant $c$ to lower bound the $\mathcal{I}(X, Y')$.
>
> Q3. *it seems to me that these findings agree with the intuitive understanding of UNMT that was already present in the UNMTs literature,*
> - We agree that some of our findings agree with other previous observations; however to the best of our knowledge, no work emphasizes the degeneration behaviors and analyses from a theoretical viewpoint the reasons of such degeneration.
> - Our analyses can shed light on the success of standard training with sound mathematical explanation, which we think can provide guidance when we encounter new demand for unsupervised translation between other modalities or domains.
> - Besides, there is one finding that is totally new, that is: the DAE regularization is to not only learn the target language model, but also function as preserving mutual information and provides the model with **initial word-to-word translation capability**. The later two functions are **much more important** than learning the target language model. This is proved in the experiment briefly mentioned in the first paragraph of Sec.3.4.2. **That is, to randomly shuffle the target word order for DAE training, the overall performance only degrades from 33 to 31.**

---

### Official Review · AnonReviewer2 · 2020-11-02

**Rating:** 5
**Confidence:** 4

**Review:**

This paper performs an ablative study on the two components involved in training unsupervised MT systems: 1) back-translation loss, 2) denoising autoencoding loss. It links the reconstruction loss to ELBO (where the q distribution is a back-translation model). It shows that the original loss with both the components is important for unsupervised MT and ELBO needs to be augmented with denoising autoencoding loss to be effective at training unsupervised MT models.

-- The graphs show comparisons on ELBO across different models. However, ELBO is not really comparable across models by definition.

-- Many findings in the paper are unsurprising and add little to our current understanding of unsupervised MT systems. For example, DAE-only collapsing to copying and BT-only collapsing to language modeling is rather expected. This can be followed from the proposed conceptualization of mutual information in the paper. The MI is highest when Y' = X! Therefore, unregularized training (with just one of the loss components) is expected to result in degenerate behavior.

-- The only slightly surprising results is that ELBO on its own collapses and fails to learn anything. Technically, it does have the two components in its loss functions that make unsupervised MT work. The reasons could range from optimization issues to poor Monte-Carlo sample based approximation of expectations in ELBO (reparametrization trick or score matching is not explored in this paper for better approximation).

---

> ### Author Response · Authors · 2020-11-23
> **We want to highlight that: DAE matters for emerging alignment knowledge of the UNMT model trained without any explicit bilingual knowledge. This surprises us most.**
>
> We sincerely thank you for your critics. We pick up some of your most concerned points and try to clarify them a little bit. Hope those response will change part of your view on our **unsurprising** findings.
>
> Q1. *However, ELBO is not really comparable across models by definition.*
> - Actually, we follow the ELBO comparison methodology along [2], [1] for computing and comparing the terms within ELBO. The model architectures are the same, and the only difference is the training protocol. We think the ELBO values are comparable indeed. Please point out what we are missing here if you insist your opinion, thanks a lot.
>
> Q2. *The MI is highest when $Y'=X$.*
> - Yes of course, since the model learns almost **copying** the input $X$ into the predicted $Y'$. One of the most surprising finding of this paper is that: **although** DAE-only training will degenerate to copy, it can also be thought of as **pre-training** prepared for BT-only training.
> As shown in the 3rd and 4th rows of Table 3 of our paper, if we conduct BT-only training after DAE-only training, we can get comparable and sometimes even better performance with standard training. That is, DAE-only is **not just** learning copy but something more. In Table 4 (the 4th row), we use ***constrained decoding*** to force the model to decode tokens that **only come from the target vocabulary**, and compute the Mutual Information (MI) between $(X, Y')$. We find that the  MI is **still very high**, which is close to the copy case. This indicates that the model is learning some **initial bilingual knowledge** as word-to-word translation even the language model fails to order them to obtain high BLEU values (as shown in the 3rd row of Table 3).
> - Another experiments which we do not included into current paper (due to space limit) is that, by using constrained decoding, we can obtain up-to 35% 1-gram precision (or BLEU-1) **after** DAE-only training compared to less than 5% before DAE-only training.
> - To conclude, we are conveying a rather **new understanding** that: DAE loss can bring **initial alignment knowledge** or basic **word translation capability** even trained on monolingual corpus, and its role in standard training is **more than** training the language model capability of the UNMT model.
>
> Q3. *The reasons could range from optimization issues to poor Monte-Carlo sample based approximation of expectations in ELBO.*
> - Firstly, currently we have only tried the REINFORCE implementation for gradient approximation, we set different sample sizes from the $q_\phi(y \vert x)$ model to approximate expectation and the trends of the learning curves turned out to be the same.
> - As we have mentioned in the last paragraph of Sec.2.2, we are not the first to make the connection to ELBO optimization, [2] has done it first to our knowledge. In their work, they try solely using the ELBO objective without DAE loss (which they dubbed as reconstruction error) and try all mainstream (more stable) gradient approximations, namely REINFORCE, **Gumbel-Softmax** and Stop-gradient, and find **all of them failed** to train the model. We recommend the reviewer to read that paper for more discussion.
> - Secondly, yes, of course we agree with you that a better approximation method might change the failure of ELBO-only training protocol into a success, but currently, we think we are still searching for it. And ELBO optimization really can lead to failure cases when **not carefully regularized**. This is what we want to emphasize with the ELBO analogy: **regularization like DAE really matters** and is task-dependent.
>
> **Reference**
>
> [1]. *Generating Sentences from a Continuous Space*, CoNLL 2016.
>
> [2]. *A probabilistic formulation of  unsupervised text style transfer*, ICLR 2020.

---

### Decision · Program_Chairs · 2021-01-07
**Final Decision**

**Decision:**

Reject

**Comment:**

This paper attempts to explain why popular UNMT training objective components (back-translation and denoising autoencoding) are effective. The paper provides experimental analysis and draws connections with ELBO and mutual information. Reviewers generally agree that the paper's goal is worthy: trying to form a better theoretical understanding of successful approaches to UNMT.
However, most reviewers raised serious concerns about the current draft and suggested another round of revision and resubmission. Specifically, reviewers were concerned that some of the analogies used to explain UNMT are underdeveloped. Further, reviewers pointed to issues with clarity that made some of the arguments hard to follow. Finally, one reviewer argued that many of the results are expected and agree with common understanding of UNMT in the literature, thus undermining their value to some extent. I tend to agree with reviewers that this paper is not ready for publication in its current form. Thus I recommend rejection.